# Heterospecific eavesdropping on an anti-parasitic referential alarm call

Shelby L. Lawson [1✉], Janice K. Enos[1], Niko C. Mendes[1], Sharon A. Gill[2] & Mark E. Hauber [1]

Referential alarm calls occur across taxa to warn of specific predator types. However, referential calls may also denote other types of dangers. Yellow warblers (*Setophaga petechia*) produce "seet" calls specifically to warn conspecifics of obligate brood parasitic brown-headed cowbirds (*Molothrus ater*), which lay their eggs in the warblers' and other species' nests. Sympatric hosts of cowbirds that do not have referential alarm calls may eavesdrop on the yellow warbler's seet call as a warning system for brood parasites. Using playback presentations, we found that red-winged blackbirds (*Agelaius phoeniceus*) eavesdrop on seet calls of yellow warblers, and respond as much to seet calls as to cowbird chatters and predator calls. Red-winged blackbirds appear to eavesdrop on seets as warning system to boost frontline defenses on their territories, although they do not seem to perceive the warblers' seets as a cue for parasitism per se, but rather for general danger to the nest.

[1] Department of Evolution, Ecology, and Behavior, School of Biological Sciences, University of Illinois at Urbana-Champaign, Urbana, IL 61801, USA.
[2] Department of Biological Sciences, Western Michigan University, Kalamazoo, MI 49008, USA. ✉email: slawson3@illinois.edu

Vocal communication involves information transfer through auditory cues from a sender to receiver[1]. Acoustic and other non-private signals can be eavesdropped upon by unintended receivers, both conspecific or heterospecific[2]. Heterospecific eavesdropping on vocal signals is common across many species of birds and mammals[3], and can provide benefits such as the earlier detection of predators, increased foraging opportunities, and better informed decision-making for habitat selection or predator avoidance[3–5]. Eavesdropping on heterospecifics can provide more or different types of information than conspecific eavesdropping alone because interspecific differences in sensory abilities and in space use within habitats likely expand the sensory-space covered by a single species[6,7].

Eavesdropping on heterospecific alarm calls, or vocalizations that alert others of nearby predation risk[8], has been detected across diverse lineages of birds and mammals[3], and provides eavesdroppers with general information about predatory threats. Referential alarm calls in particular can indicate which of a suite of threats is present, each requiring specific actions to evade[9–15]. Referential alarm calls are vocalizations that denote specific objects, and elicit particular behavioral responses from animals that hear these calls[9]. For example, Verreaux's sifakas (*Propithecus verreauxi*) produce different referential alarm calls for aerial vs. terrestrial predators, and listeners respond with specific anti-predatory responses depending on the predator type referenced[16], which likely increases survival and imparts fitness benefits[17].

Con- and heterospecific eavesdropping on referential alarm calling is particularly widespread in songbirds, and is well-studied in the context of improving nest defense and minimizing nest detection[18–24]. In turn, threats to nests include both nest predators that depredate eggs and nestlings, and brood parasites that forgo nest building and instead lay their eggs in the nests of others, leaving the host adults to care for the foreign egg(s) and chick(s)[25]. To protect the nest from these threats, many species act aggressively toward both predatory and parasitic intruders within their territories/near their nests[26–29].

In the context of anti-parasitic nest defense, success hinges on early detection of brood parasites prior to their discovery of the host's nest[30]. Eavesdropping on alarm calls that signal brood parasites could have several possible benefits, including increased individual vigilance, social mobbing, and, eventually, decreased parasitism, especially for species that do not have a referential system of their own. Therefore, it is hypothesized that eavesdropping on heterospecific referential alarm calls that signal brood parasitic threats should evolve. Prior work has identified parasite-specific alarm calls in diverse host–parasite systems[29,31], and as predicted, the first evidence for heterospecific recognition of these calls was recently described across two species of congeneric *Acrocephalus* reed warbler hosts of the common cuckoo (*Cuculus canorus*) that use acoustically similar calls to signal the parasite's presence[32]. However, none of these calls fit the full definition of referential alarm calls, as their elicited responses include general recruitment and mobbing, but not specifically anti-parasitic defenses.

The yellow warbler (*Setophaga petechia*) is the only known species to use referential alarm calls to signal the presence of a brood parasite[19]. In response to obligate brood parasitic brown-headed cowbirds (*Molothrus ater*, hereafter "cowbirds"), yellow warblers of both sexes produce a "seet call" that alerts their pair-bonded mates of the brood parasitic threat[33], and, uniquely, females do not recruit to mob the parasitic threat but instead return to and sit tightly on their nest, therefore physically reducing brood parasitism risk. Seet calls are only produced in response to cowbirds, and almost exclusively during laying and incubation when the nest is at the highest risk of parasitism[19,30,33]. Yellow warblers thus present an exceptional system to study whether any other hosts of brood parasites eavesdrop on heterospecific referential alarm calls to boost their own nest defenses toward cowbirds.

Here we present new evidence that red-winged blackbirds (*Agelaius phoeniceus*, hereafter "redwings"), another common host of brown-headed cowbirds[34–36] which are phylogenetically distant and vocally distinct from yellow warblers, actively eavesdrop upon and respond to seet calls, and may thus potentially use neighboring yellow warbler calls as an early warning system against brood parasitism risk. Redwings often nest in loose aggregations with yellow warblers where closer proximity to redwing nests is correlated with a decrease in parasitism rates of the nearby warblers[37]. This may be because redwings frontload their nest defenses, meaning that they use both vocal and physical aggression toward cowbirds to prevent them from accessing and parasitizing the nest[33,38–46]. However, redwings do not have a known referential alarm call system of their own, and as such, redwings may eavesdrop on yellow warblers' "seet calls" to enhance their own nest defense against cowbirds.

To assess whether redwings use yellow warbler seet calls as a frontline defense against cowbird parasitism, we report on three analyses from two playback experiments. The first experiment was conducted on yellow warbler territories for a separate study, but that also comprised heterospecific (including redwing) data. The second experiment sought to directly test redwings' responses to the playback types. In each experiment, we broadcast seet calls, cowbird, and nest predator vocalizations, as well as procedural controls at focal individuals. For the first experiment, we broadcast playback treatments on yellow warbler territories and tested the hypothesis that redwings respond similarly to cowbird chatters and warbler seet calls, but not to the other warbler alarm call (the generic "chip") or other heterospecific vocalizations from neighboring territories. For the second study, we located active redwing nests to investigate the hypothesis that territorial redwings respond to seet calls to enhance their frontloaded nest defenses against cowbirds. We predicted again that redwings' responses to seet calls would be similar to cowbird chatters, but not to other heterospecific calls. For the final analyses, we were interested whether in the second experiment, the redwings' distance to the closest yellow warbler territory influenced redwing responses (specifically, calling rates) toward playbacks of cowbird chatters and seet calls. If redwings actively eavesdrop on seet calls as a warning signal for brood parasitism, and seet calls are perceptually equivalent to chatters, then redwings nesting closer to yellow warbler territories are predicted to mount stronger responses (more calls) to playbacks of cowbird chatters and seet calls than redwings that nest farther away from yellow warblers and thus do not have access to the heterospecific hosts' signal.

## Results

**Experiment 1: playbacks on yellow warbler territories**. Male redwing presence on yellow warbler territories differed significantly by playback treatment ($\chi^2 = 17.08$, $p < 0.001$; Fig. 1). Based on post hoc analyses, we found that redwings were present on yellow warbler territories more often during seet calls trials (41%) compared with both chip calls (19%, $p = 0.037$) and wood thrush song (*Hylocichla mustelina*) controls (3%, $p < 0.001$). In contrast, redwing presence on yellow warbler territories did not differ between seet calls and cowbird chatters (46%, $p = 0.189$).

## Experiment 2: playback at red-winged blackbird nests

**Latency**. Average latencies (including zeros) to respond to the treatments were highly variable for both redwing males (Fig. 2)

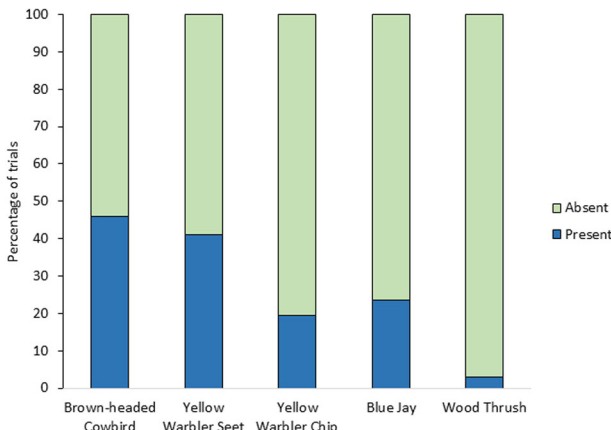

**Fig. 1 Percentage of trials for different playback treatments on yellow warbler territories in which at least one male redwing was present and responsive to the playback (dark blue) or absent (light green).** Data were analyzed using a nominal logistic regression. Playbacks include cowbird chatters ($n = 37$), warbler seets ($n = 34$), warbler chips ($n = 31$), blue jay calls ($n = 34$), and wood thrush songs ($n = 35$).

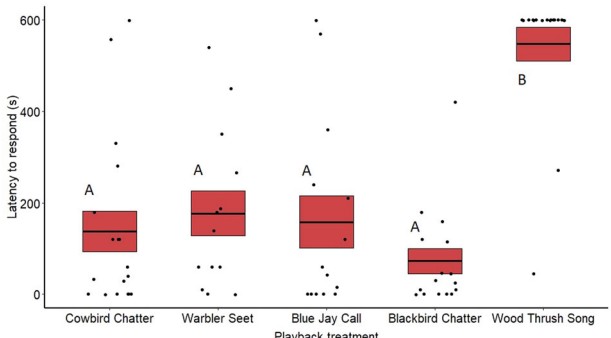

**Fig. 3 Latency for redwing females to respond to the playback treatments at redwing nests.** Means are shown with the bold line, and shaded boxes represent standard errors. Data were analyzed using a zero-inflated negative binomial model. Boxes with different letters denote post hoc statistical differences between treatments (cowbird chatters $n = 18$, warbler seets $n = 13$, redwing chatters $n = 16$, blue jay calls $n = 16$, wood thrush songs $n = 17$). For $p$-values of post hoc comparisons, please refer to Supplementary Table 1.

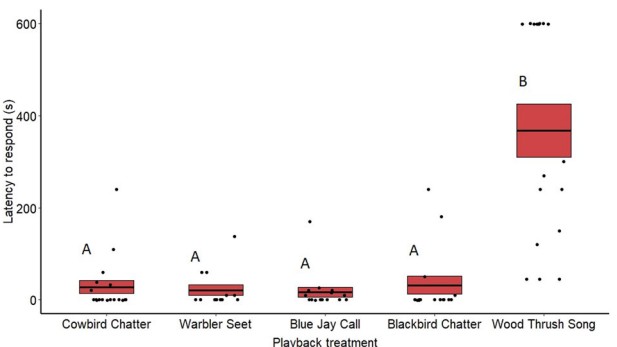

**Fig. 2 Latency for redwing males to respond to the playback treatments on redwing territories and at nests.** Means are shown with the bold line, and shaded boxes represent standard errors. Data were analyzed using a zero-inflated negative binomial model. Boxes with different letters denote post hoc statistical differences between treatments (cowbird chatters $n = 23$, warbler seets $n = 18$, redwing chatters $n = 25$, blue jay calls $n = 16$, wood thrush $n = 23$). For $p$-values of post hoc comparisons, please refer to Supplementary Table 1.

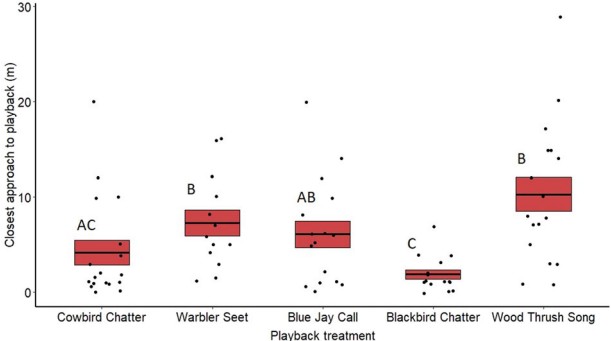

**Fig. 4 Closest approach to the playback speaker (in meters) by redwing males for the different treatments on redwing territories and at nests.** Means are shown with the bold line, and shaded boxes represent standard errors. Data were analyzed using a general linear mixed model. Boxes with different letters denote post hoc statistical differences between treatments (cowbird chatters $n = 23$, warbler seets $n = 18$, redwing chatters $n = 25$, blue jay calls $n = 16$, wood thrush songs $n = 23$). For $p$-values of post hoc comparisons, please refer to Supplementary Table 2.

and females (Fig. 3). The ratio of trials where male redwings responded immediately (latency of zero) to the playback differed significantly between treatments ($N = 104$, $\chi^2 = 26.06$, $p < 0.001$). Based on post hoc pairwise comparisons of least-squares means, males were more likely to respond immediately to playbacks of cowbird chatters ($p < 0.001$), seet calls ($p < 0.001$), blue jay calls (*Cyanocitta cristata*, $p < 0.001$), and redwing chatters ($p < 0.001$), compared with wood thrush (control) songs. The ratio of trials where redwing females responded immediately (latency of zero) to the playback did not differ significantly between treatments ($N = 77$, $\chi^2 = 7.23$, $p = 0.123$). Nonzero latencies also differed significantly between treatments for males ($F_{4,56} = 16.37$, $p < 0.001$) and females ($F_{4,59} = 16.13$, $p < 0.0001$). Based on post hoc pairwise comparisons of least-squares means, male redwings responded more quickly to cowbird chatters ($z = 5.12$, $p < 0.001$), seet calls ($z = 5.59$, $p < 0.001$), blue jay calls ($z = 5.18$, $p < 0.001$), and redwing chatters ($z = 6.25$, $p < 0.001$) compared with wood thrush songs. Female latencies showed the same pattern, where females responded more quickly to playbacks of cowbird chatters ($z = 5.72$, $p < 0.001$), seet calls ($z = 5.32$, $p < 0.001$), blue jay calls

($z = 4.54$, $p < 0.001$), and redwing chatters ($z = 7.45$, $p < 0.001$) compared with wood thrush songs. Latency to respond did not differ between any of the non-control playbacks for either sex (see Supplementary Table 1 for nonsignificant comparisons). Date of playback did not have a significant effect on nonzero latencies for males ($F_{1,103} = 0.372$, $p = 0.54$) or females ($F_{1,77} = 2.71$, $p = 0.108$) nor did year (males only) ($F_{1,56} = 0.36$, $p = 0.54$).

**Closest approach.** Closest approach also varied significantly between the treatments for both males ($F_{4,104} = 10.25$, $p < 0.001$) (Fig. 4) and females ($F_{4,77} = 3.68$, $p < 0.01$) (Fig. 5). Redwing males moved significantly closer to the speaker during redwing chatter playbacks compared with seets ($z = 4.18$, $p < 0.001$), blue jay calls ($z = -2.84$, $p = 0.03$) and wood thrush songs ($z = 5.81$, $p < 0.001$). Redwing males also approached the speaker more closely during cowbird playbacks than seets ($z = 2.90$, $p = 0.02$) and wood thrush songs ($z = 4.43$, $p < 0.001$). Females only approached more closely to playbacks of redwing chatters compared with the control wood thrush songs ($z = 3.57$, $p < 0.01$). All other pairwise comparisons were not significantly different (see Supplementary Table 2). Date of

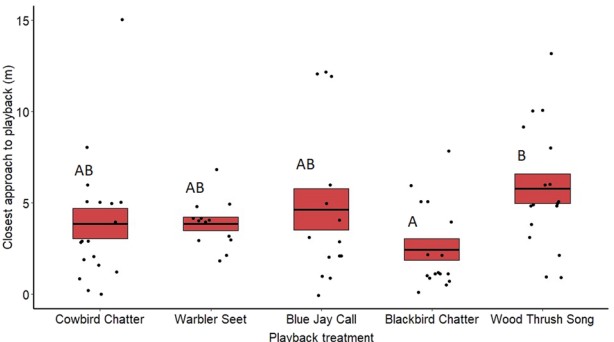

**Fig. 5 Closest approach to playback speaker by redwing females to respond to the playback treatments at redwing nests.** Means are shown with the bold line, and shaded boxes represent standard errors. Data were analyzed using a general linear mixed model. Boxes with different letters denote post hoc statistical differences between treatments (cowbird chatters $n = 18$, warbler seets $n = 13$, redwing chatters $n = 16$, blue jay calls $n = 16$, wood thrush songs $n = 17$). For $p$-values of post hoc comparisons, please refer to Supplementary Table 2.

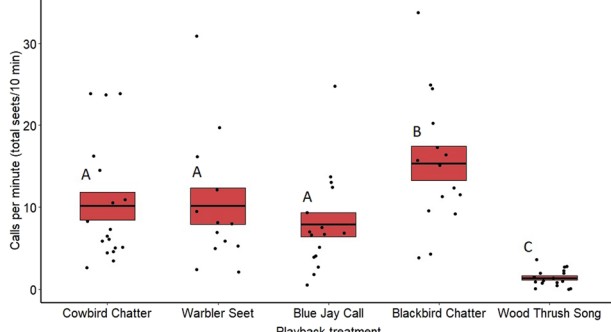

**Fig. 7 Call rate for redwing males in each treatment in 2019 at redwing nests.** Means are shown with the bold line, and shaded boxes represent standard errors. Data were analyzed using a general linear mixed model. Boxes with different letters denote post hoc statistical differences between treatments (cowbird chatters $n = 18$, warbler seets $n = 13$, redwing chatters $n = 16$, blue jay calls $n = 16$, wood thrus songs $n = 17$). For $p$-values of post hoc comparisons, please refer to Supplementary Table 3.

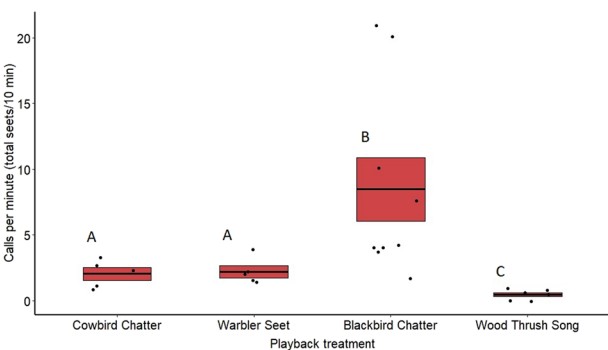

**Fig. 6 Average call rate for redwing males for treatments in 2018 on redwing territories.** Means are shown with the bold line, and shaded boxes represent standard errors. Boxes with different letters denote post hoc statistical differences. Data were analyzed using a general linear mixed model. Boxes with different letters denote post hoc statistical differences between treatments (cowbird chatters $n = 5$, warbler seets $n = 5$, redwing chatters $n = 9$, wood thrush songs $n = 6$). For $p$-values of post hoc comparisons, please refer to Supplementary Table 3.

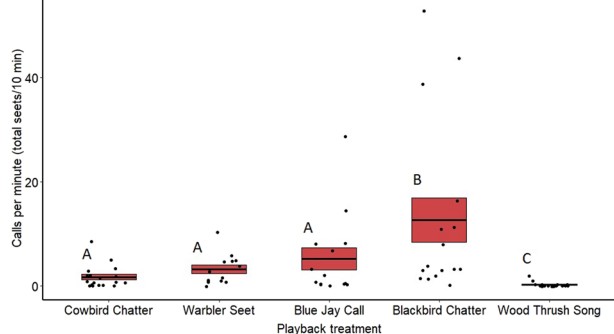

**Fig. 8 Call rate for redwing females for playback treatments at redwing nests.** Means are shown with the bold line, and shaded boxes represent standard errors. Data were analyzed using a general linear mixed model. Boxes with different letters denote post hoc statistical differences between treatments (cowbird chatters $n = 18$, warbler seets $n = 13$, redwing chatters $n = 16$, blue jay calls $n = 16$, wood thrush $n = 17$). For $p$-values of post hoc comparisons, please refer to Supplementary Table 3.

playback did not have an effect on male ($F_{1,104} = 0.015$, $p = 0.90$) or female approach ($F_{1,77} = 0.65$, $p = 0.42$), and neither did year (males only) ($F_{1,104} = 0.911$, $p = 0.34$).

**Calling rate**. Rate of calling by redwing males differed significantly between treatments both in 2018 ($F_{3,24} = 16.53$, $p < 0.0001$) (Fig. 6) and 2019 ($F_{4,78} = 25.81$, $p < 0.0001$) (Fig. 7). Based on post hoc pairwise comparisons of least-squares means, in both 2018 and 2019 redwing males called more toward playbacks of cowbird chatters (2018: $z = -3.04$, $p = 0.01$; 2019: $z = -7.63$, $p < 0.001$), seet calls (2018: $z = -2.66$, $p = 0.03$; 2019: $z = -6.29$, $p < 0.001$), and redwing chatters (2018: $z = -8.16$, $p < 0.001$; 2019: $z = -9.76$, $p < 0.001$) compared with wood thrush songs. Redwings also called more toward playbacks of redwing chatter compared with cowbird chatter (2018: $z = 4.37$, $p < 0.001$; 2019: $z = 2.73$, $p = 0.04$), seet calls (2018: $z = -4.45$, $p < 0.001$; 2019: $z = -2.98$, $p = 0.02$), and in blue jay calls (2019: $z = 4.35$, $p < 0.001$). Calling rate did not differ significantly between cowbird and seet treatments in either 2018 ($z = -0.27$, $p = 0.99$) or 2019 ($z = -0.44$, $p = 0.99$) (see Supplementary Table 3). Date of playback had a significant effect on male calling

rate in 2018 ($F_{1,24} = 9.17$, $p = 0.01$), where calling rate increased later in the season, however, this was not seen in 2019 ($F_{1,78} = 1.78$, $p = 0.18$).

Female calling rate also differed significantly between treatments ($F_{4,77} = 13.73$, $p < 0.0001$) (Fig. 8). Similar to males, females called more toward playbacks of cowbird chatters ($z = -2.81$, $p = 0.039$), seet calls ($z = -4.42$, $p < 0.001$), blue jay calls ($z = -4.41$, $p < 0.001$), and redwing chatters ($z = -7.28$, $p < 0.001$) compared with wood thrush songs. Females also called more toward playbacks of redwing chatters compared with blue jay calls ($z = 2.76$, $p = 0.044$). Unlike males, calling rate for females was similar between redwing chatters and seet playbacks ($z = -2.45$, $p = 0.10$), and females called more during redwing chatter playbacks than cowbird chatters ($z = 4.69$, $p < 0.001$). Notably, females also did not differ in calling rate between the seet and cowbird treatments ($z = 1.99$, $p = 0.26$) (see Supplementary Table 3). Date of playback ($F_{1,77} = 0.30$, $p = 0.58$) did not have a significant effect on female calling rate.

**Calling rate across distances between heterospecific territories.** For male redwings, distance to nearest yellow warbler territory had a significantly negative effect on alarm-calling rate during cowbird

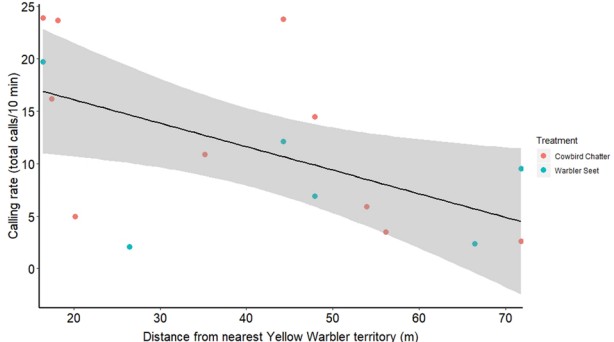

**Fig. 9 Calling rate of male redwings during seet and cowbird playbacks at redwing nests (2019) plotted along distance from the focal pair's nest to nearest yellow warbler territory ($R^2 = 0.26$, $F_{1,14} = 6.38$, $p = 0.02$).** Gray areas represent the 95% confidence interval of the slope. Treatments are marked with orange (cowbird) or blue (seet) coloring.

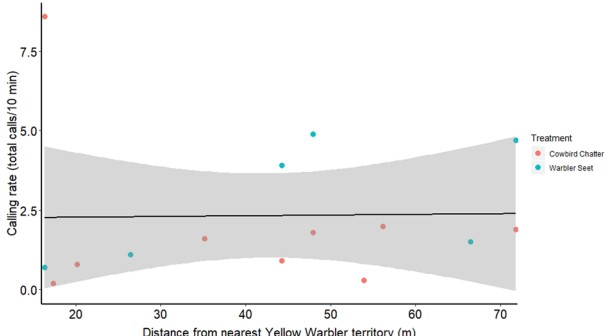

**Fig. 10 Calling rate of female redwings during seet and cowbird playbacks at redwing nests (2019) plotted along distance from the focal pair's nest to nearest yellow warbler territory ($F_{1,13} = 0.25$, $p = 0.62$).** Treatments are marked with orange (cowbird) or blue (seet) coloring.

and seet playbacks ($R^2 = 0.24$, $F_{1,14} = 5.32$, $p = 0.03$; Fig. 9): male redwings with nests further away from the nearest yellow warbler territory called less toward playbacks of seets and cowbird chatters. Playback treatment itself was not a significant predictor, in that the focal redwings' alarm responses to both cowbird chatters and seets showed a similarly negative pattern of nest/territory-distance dependence ($F_{1,14} = 0.51$, $p = 0.48$). We did not see this pattern for female response rates based on distance ($F_{1,13} < 0.001$, $p = 0.98$; Fig. 10) or treatment ($F_{1,13} = 0.37$, $p = 0.55$).

## Discussion

Our results demonstrate a series of patterns in which (1) both male and female redwings respond to the seet call as to other signals for danger to their nests, (2) both sexes of redwings show response equivalency between cowbird chatters and seet calls, (3) male redwings respond to anti-parasitic yellow warbler calls similarly to cowbird chatters, but not to yellow warblers chip calls, and (4) male redwings respond more to cowbird chatters and seet calls coming from closer yellow warbler neighbors. Taken together, these lines of evidence support that redwings eavesdrop on heterospecific referential alarm calls to enhance frontline defenses against parasitic cowbirds and other threats to their own nest.

In our first experiment, we found that nearly half of all cowbird chatters and seet call playbacks on yellow warbler territories drew in neighboring redwings. Redwings were as likely to respond to seet playbacks as they were to cowbird playbacks but, critically, redwings responded to a significantly higher proportion of seet

playbacks compared with chip playbacks. Our results suggest that redwing males selectively respond to the seet calls of yellow warbler neighbors as much as they do to nearby cowbirds, and more so than chip calls. For yellow warblers, seet calls warn against brood parasites, while chip calls warn of general predatory threats both to the adult and to the nest[18,47]. Differential redwing responses to the calls suggest that this host species is eavesdropping on and possibly discriminating between the two heterospecific warbler call types. Thus, it may be possible that redwings have a cognitive algorithm to discern between information in yellow warbler seet and chip calls, although this requires further neurofunctional tests.

In the second experiment, we found that when playbacks were presented on redwing territories, both males and females responded to seet calls with similar aggression (in terms of latency, alarm calling and closest approach) to cowbird chatters and blue jay calls, and both less so than to redwing chatters, indicating redwings treat seet calls with the same urgency as calls that simulate threats to breeding success, regardless of sex. Therefore, redwings appear to perceive seet calls as an indicator of a nest threat present, but not necessarily as a referential alarm call specifically informing brood parasitism threat, because responses to seets and Blue Jay calls did not differ. In turn, redwing females, unlike yellow warblers, appear to lack the nest approach and sitting response when hearing cowbird chatters. The aggressive responses we observed in response to seet calls are likely a generic nest-protection response by redwings, as recorded by our observations. Redwings also responded to redwing chatter with the most aggression, suggesting that they perceive conspecific intruders as the gravest threat among our simulated set of danger cues.

In the third analysis, we found that male vocal aggression toward cowbird and seet calls was negatively correlated with distance from nearest yellow warbler territory. This pattern, however, was not seen in females. These results suggest a "neighborhood watch" effect, in that redwing males nesting closer to yellow warblers may be more sensitive to the latter species' referential seet calls (and the presence of cowbirds through their calls). As such, red wings who have more access to the seet warning calls mount stronger frontline defenses to cowbirds and the seets compared with redwings that have access to this information. In addition, there was no difference between male redwings' own alarm responses to playbacks of seets vs. cowbird chatters, suggesting that redwings closer to yellow warblers are more responsive to both the warblers' referential calls and the cowbirds' own vocalizations that they reference. Greater redwing mobbing through stronger responses to yellow warbler anti-parasitic alarm calls may also contribute to the observed higher nesting success of yellow warblers nearer to redwing nests[37]. Redwings frontload their nest defenses against brood parasitism[33,38–45], so the ability to respond specifically to anti-parasitic seet calls as a warning for cowbirds could impart fitness benefits to redwings that eavesdrop on the call. This, along with the often close nesting proximity between redwing and yellow warbler nests at wetland sites[37,48,49] (and pers. obs.), may have primed redwings evolutionarily to pay attention to the warbler seet call.

There are other wetland species that may potentially eavesdrop on seet calls, but we do not see evidence of this in our data. Specifically, when we examined the responses of species other than yellow warblers and redwings during these same set of playbacks at yellow warbler territories (see Supplementary Table 4), we found an opposite pattern, whereby other locally breeding species responded more to the yellow warbler's chip playbacks (20/31 trials, 12 species combined) than to seet calls (8/34 trials, 7 species combined) (Fisher's exact test, $p < 0.05$). Furthermore, while some of the species responded several times to

different chip trials, only one species ever appeared in more than one seet trial. Hence, eavesdropping on heterospecific seet calls as a warning signal may be an attribute unique to redwings at our study sites.

From our data, it appears redwings actively eavesdrop on the seet calls of nearby yellow warblers and respond aggressively when exposed to seet calls, both on their own territories as well as neighboring yellow warbler territories. However, we found general behavioral equivalency in the strengths of the responses not only to seet calls and cowbird chatters, but also to blue jay calls (Experiment 2, Fig. 4), suggesting that redwings may perceive seets as a general alarm call rather than a referential alarm call for cowbird. Nonetheless, on yellow warbler territories redwings did respond more to cowbird and seet playbacks compared chip playbacks, suggesting there may be some discrimination between heterospecific general alarm calls and anti-parasitic referential calls, but this needs further testing.

A limitation of our study is that data were only collected during incubation, when the risk of brood parasitism is highest. In the future, conducting playbacks across different nesting stages may help us better understand whether redwings can respond to the referential meaning of the yellow warbler's seet call. Prior work[41] demonstrated that during incubation stage, redwings respond more to cowbird models than models of a nest predator. After the eggs hatch and brood parasitism is no longer a threat, responses to cowbird models decrease, while aggression toward nest predator models increase with increasing investment in young. If redwings are using seet calls as frontline nest defense specifically against brood parasitism, we would expect aggressive responses to seet calls to be strongest during laying and incubation and weaker after the eggs hatch.

In conclusion, our results demonstrate that redwings actively eavesdrop on and respond to yellow warbler seets as a frontline defense to protect their nest investment. Redwings do not appear to have a referential system of their own and may instead eavesdrop on the yellow warbler's seet call to use it for their benefit. In addition, redwings that nest near yellow warblers respond more strongly to parasitic cues. Given that redwing proximity has been shown to reduce the probability of brood parasitism by cowbirds upon yellow warbler nests nearby[37], our own findings open up questions for future research to explore whether yellow warblers and redwings possibly have a mutualistic communicative relationship, whereby yellow warblers are the alarm system providing warning cues for cowbird presence, and redwings are the aggressors keeping cowbirds at bay.

## Methods

**Sites and study species**. Both playback experiments (described below) took place in multiple wetlands in Champaign ($n = 3$), Iroquois ($n = 1$), and Vermilion counties ($n = 3$) in central Illinois, USA. Sites were comprised of mesic shrubland habitat, with dominant shrubs including willow (*Salix* spp.), dogwood (*Cornus* spp.), and Autumn Olive (*Elaeagnus umbellate*), with mesic grasses abundant among shrubs. Patches of cattails (*Typha* spp.) and reed (*Phragmites* spp.) were prevalent along bodies of water at most sites[50,51].

Redwings are sympatric with yellow warblers in Illinois, and both are parasitised by cowbirds[48,49,52] (pers. obs.). At our sites, redwings arrive as early as February and nest from late-April through late-July, with peak breeding in late-May[50,51] (pers. obs). Redwings are polygynous, and males may have several nests from different females on their territory[34]. Yellow warblers arrive at our sites late-April and breed from early-May through late June, with peak breeding mid-to-late-May. At these sites the interspecific overlap of territories was common between redwings and yellow warblers (pers. obs.).

**Playback stimuli construction**. For our experiments, we constructed playlists for six different playback treatments: (1) female cowbird chatters (brood parasite), (2) yellow warbler seet calls (cowbird-specific anti-parasitic alarm call[17–19,30,42]), (3) yellow warbler chip calls (general antipredatory alarm call[47,53]), (4) redwing chatters (general conspecific vocalization[54]), (5) blue jay calls (a warbler and redwing nest predator[55]), and (6) wood thrush songs (an innocuous heterospecific

control that is sympatric with redwings but do not prey, parasitize, or compete with them[56]). We included blue jay calls to tease apart if redwings responded to seet calls as a general alarm call, or a referential alarm call specifically informing brood parasitism risk. Using both a brood parasite and predator call presentation is necessary to fully understand whether the host's aggressive responses are specifically anti-parasitic or general[57]. The chip was chosen as a relevant general alarm stimulus for playbacks on yellow warbler territories, and redwing chatters were used as a relevant territory intrusion stimulus on redwing territories. Note that experiments 1 and 2 differed in which treatments were used (described below).

We used audio files from Xeno Canto, all sourced from the Midwestern and Southwestern United States (Colorado, Illinois, Michigan, Minnesota, and Ohio), except for seet calls, which were sourced directly from Gill and Sealy[18], and redwing chatters, which were sourced from Lynch et al.[58]. Playlists were created using Adobe Audition CC 2019. To avoid pseudoreplication[59], we constructed five different playlist files for each treatment, and chose one exemplar file randomly for each playback trial (described below). Each playlist was comprised of vocalizations from at least three different individuals. Vocalizations from individuals were placed in a random order, and then repeated to obtain the 10-min playlist. Intervals of silence were placed into between vocalizations, with intervals ranging from 2 to 6 s based on rates found in natural recordings on Xeno Canto. Amplitude was adjusted such that sounds played from our speaker at full volume were ~90 dB (measured 0.5 m from speaker). To minimize signal-to-noise ratio in playback files, frequencies below 500 Hz, which are well below the range of any of our stimuli, were filtered out.

For both experiments, playbacks were conducted with an AYL-SoundFit speaker connected to a Samsung Galaxy 8 cellular phone with the audio files. We placed equipment ~1 m high in vegetation and recorded data from > 10 m away. Playback trials occurred for 10 min. For both experiments, we retested each territory 24–72 h later (mean = 41) to avoid habituation, with a different, randomly assigned treatment to prevent order effects. All statistical tests were conducted in the statistical program R 3.6.1 (packages lme4, nlme, multcomp, emmeans and car; see "Statistical analyses" section), with $\alpha = 0.05$.

These studies were approved by the animal ethics committee (IACUC) of the University of Illinois (#17259), and by USA federal (MB08861A-3) and Illinois state agencies (NH19.6279).

**Experiment 1: playback on yellow warbler territories**. Playback experiment: To assess if redwings respond similarly to cowbird chatters and seet calls, but not to other yellow warbler or heterospecific calls, we first used data collected during playback trials at active yellow warbler territories. Warbler territories were determined to be active in two ways: (1) if a nest with eggs was found on the territory, (2) if a nest could not be found but both a male and female were present on the territory on checks across multiple days and the pair produced alarm calls at the experimenter, which is indicative of nesting investment on the territory[54] (pers. obs.). We also excluded any yellow warbler pairs seen carrying nesting material or insects, which signify building and nestling stage, respectively. Seet calls are almost exclusively produced during laying and incubation stage[33,47], thus we only presented playbacks on territories presumed to be in those stages. Yellow warbler playbacks trials occurred from mid-May to late June in 2018 and 2019, and between 0500 and 1200 h local time. We did not systematically band territorial birds at our sites for individual identification prior to experimentation. Therefore, all nests tested were ≥ 30 m apart to maintain independence, as nests at this distance likely belong to different breeding units based on average territory size[50,60,61]. In addition, we waited 30–60 min between playbacks at neighboring sites to avoid any carryover effects on neighbors.

Yellow warbler territories received one of five different playback treatments on two separate days, such that each territory was tested with two of the five playback types. Playbacks included cowbird chatters ($n = 37$), yellow warbler seets ($n = 34$), yellow warbler chips ($n = 31$), blue jay calls ($n = 34$), and wood thrush songs as an innocuous heterospecific control ($n = 35$), for a total of 171 playbacks. The playback speaker was placed 5–6 m from the yellow warbler male's commonly used song post. During this set of initial playback trials, we specifically recorded a single binary response variable of whether any redwings responded to the playback or not. A response was only marked if one or more male redwings were present and alarm calling within 30 m (radial average distance of territory size[56]) of the speaker any time during the 10-min playback. Redwing alarm calls are described in Knight and Temple[54]. We also report on which other species, apart from yellow warblers, responded to these same playback types during this experiment (see Supplementary Table 4).

**Experiment 2: playback on red-winged blackbird territories and at nests**. Playback experiment: To investigate if territorial redwings respond to seet calls to enhance their frontloaded nest defenses against cowbirds, we conducted playback trials at active redwing nests in 2018 and 2019. Playbacks were conducted using the same methodology and site locations in Champaign and Vermillion counties as Experiment 1. In Experiment 2, we increased the distance at which nests/song posts were tested, to maintain independence, to ≥50 m apart, as average territory size is larger for redwings than yellow warblers[34]. This distance also prevented us from inadvertently testing the same parents twice at different nests, as redwings are polygynous harem breeders within their territories[34]. Playbacks in 2018 were conducted 5 m from the male's focal song post. We were unable to search for nests

in 2018, but used behavioral observations to select pairs that likely had an active nest (e.g., alarm calling, no nest material carried or fledglings present). In 2019, we instead placed speakers 5 m from known, located active nests and conducted playbacks during the incubation stage, as this is when cowbirds pose the gravest threat[62]. In 2019, we conducted playbacks with speakers 5 m from focal nests, instead of song posts to reliably simulate the threats to the nest. We thoroughly searched sites 1–2 times weekly for active nests. Nest contents were checked every 3 days to ensure playback trials occurred during incubation.

Similar to experiment 1, redwings received one of five different playback treatments on 2 separate days, such that each pair was tested with two of the five playbacks: cowbird chatters ($n = 5$ in 2018; $n = 18$ in 2019), yellow warbler seets ($n = 5$ in 2018; $n = 13$ in 2019), redwing chatters ($n = 9$ in 2018, $n = 16$ in 2019), blue jay calls ($n = 0$ in 2018; $n = 16$ in 2019), and wood thrush songs ($n = 6$ in 2018; $n = 17$ in 2019), for a total of 105 playbacks across 2 years. In 2019, two nests were not retested as they were depredated between trials. For logistical reasons, we did not include blue jay calls in 2018. Only male responses were recorded in 2018 because only males responded to the playbacks on yellow warbler territories. After noting that females responded as well near known active nests, we also recorded responses for both sexes in 2019 (see "Statistics and reproducibility" below).

During the playback trial, we recorded the following behavioral responses from the target individual within 30 m of the speaker: (1) response latency (sec after the start of trial when a switch to aggressive behaviors occurred: posturing, hopping, alarm calling, or attacking the speaker); (2) closest approach to the speaker (m); and (3) call rate (total calls/10 min). In 2018, we only counted "checks" and "cheers" as these are general alarm calls used by redwings in many contexts[63]. In 2019, we expanded to counting "checks", "chits", "chonks", and "cheers" as they are all used interchangeably as nest defense alarm calls by both sexes, although only males produce cheer calls[54,63,64]. Therefore, we analyzed call rate for 2018 and 2019 separately.

**Calling rate analysis across distances between heterospecific territories**. We used spatial data to evaluate if redwings nesting closer to yellow warbler territories mount stronger responses (calls) to playbacks of cowbird chatters and seet calls than redwings nesting farther away that do not access to the heterospecific hosts' signal. In 2019, we recorded locations of playbacks conducted on yellow warbler territories, active redwing nests and yellow warbler territories using GPS units with 3 m accuracy (Garmin eTrex 10). Using software in ArcGIS (ver 10.4 ESRI), we measured the distance (m) of each redwing nest to the nearest playback conducted on a yellow warbler territory. We assumed that we found all active redwing nests and warbler territories within these study subsites. Prior work[37] found that redwings would respond to playbacks up to 60 m from the nest, which coincided with the redwings' territory boundaries. Many of our territories in marginal, upland habitats were larger than the reported average, so we extended our cutoff range to count redwing nests that were up to 75 m away from a yellow warbler territory. If redwings were deriving any benefit from eavesdropping on cowbird danger as predicted by chatters and seets, then they would need to nest within this range from a yellow warbler nest to be able to eavesdrop on and respond to their neighbor's seets.

**Statistics and reproducibility**. Experiment 1: playback on yellow warbler territories: We ran a nominal logistic regression that analyzed the effect of playback treatment on whether redwings were absent or present during trials. We then ran post hoc Tukey pairwise comparisons between cowbird vs seet treatments, seet vs wood thrush, and seet vs chip treatments to account for multiple comparisons between playback type pairs.

Experiment 2: playback on red-winged blackbird territories and at nests: We evaluated whether playback treatment affected the three response variables of interest (latency, total alarm calls, and closest approach) using a separate model for each. For latency and approach, we combined the data from 2018 and 2019, but for call rates, we analyzed data separately for the two years because some call types were not counted in 2018 and only males were recorded that year (see above). We determined if redwing (males and females separately) responded immediately (latency of <1 s) or with some latency (≥1 s) and conducted a $\chi^2$ test on the ratios by treatment to determine if birds were more likely to respond immediately during particular playback treatments. We also ran a negative binomial generalized mixed model on the nonzero response latencies (# of seconds to respond) with playback treatment and date as fixed effects, and redwing nest site ID as a random effect. For alarm call and closest approach variables, we log transformed the data after adding a small constant and ran general linear mixed models. In each model, we included playback treatment and date as fixed effects, and redwing nest site ID as a random effect. For all three models, we ran post hoc Tukey tests to multiple compare treatment pairs of least-square means.

**Calling rate analysis across distances between heterospecific territories**. We ran separate analyses of variance for males and females, which analyzed calling rate (calls per min) during seet and cowbird playbacks with distance from the focal redwing's nest to the nearest yellow warbler territory and the playback call treatment (seet vs. chatter) as fixed effects.

**Reporting summary**. Further information on research design is available in the Nature Research Reporting Summary linked to this article.

## Data availability

The data sets generated during and/or analyzed during the current study are available in the Dryad repository, https://doi.org/10.5061/dryad.zpc866t57.

## Code availability

The code used for analysis for this study is available from the corresponding author upon reasonable request.

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

## Acknowledgements

We thank the Champaign County Forest Preserve District, the Vermilion County Conservation District, and the Illinois Department of Natural Resources for permitting us to use their parks for our research. We also thank Mikus Abolins-Abols and Becky Fuller for their assistance with statistics, as well as Ken Yasukawa and Matt Louder for stimuli and discussions. We also thank www.xeno-canto.org for suppling the recordings used for playbacks. Work for this project was supported by the American Ornithological Society [to S.L.L.] the National Geographic Society [to M.E.H., NGS-60453R-19], and by the Harley Jones Van Cleave Professorship [to M.E.H.] and the School of Integrative Biology [to S.L.L., Clark Research Support Grant, Lebus Fund Award, Dissertation Travel Grant] at the University of Illinois at Urbana-Champaign.

## Author contributions

S.L.L. and M.E.H. co-conceived of the study. S.L.L., S.A.G., and M.E.H. participated in the design of the study. S.L.L., J.K.E., and N.C.M. carried out the field work and playbacks. S.L.L., J.K.E., and M.E.H. conducted the data analysis. S.L.L. drafted the paper, and all authors critically revised the paper. S.L.L. and M.E.H. obtained funding. All authors gave final approval for publication, and agree to be held accountable for the work performed therein.

## Competing interests

The authors declare no competing interests.
