## [Peer Review File · Communications Biology]

Reviewers' comments:

Reviewer #1 (Remarks to the Author):

In this paper the authors study the use of referential alarm calls in red-winged blackbirds in a heterospecific context to assess the use of information from heterospecific calls (yellow warbler) about a particular nest parasite versus more general threats from predators. The authors use a range of playback experiments to test the idea that the redwings assess the information in the call of the yellow warbler which is a heterospecific that often nests close to the red-winged blackbird nests. I think that this is an interesting idea and area of research. Generally the paper is very well written and presented. We have three major concerns about the manuscript:

1. The authors need to layout the logic of the study more clearly, in terms of:

- What pattern is expected in response to different treatments, to demonstrate that redwings are using warbler seet calls in a very specific way? This needs to be clearly spelt out in the Introduction and should guide the presentation of results and discussion.
- What is the motivation for doing experiments both in warbler territories and near redwing nests, i.e. what do the authors expect to be functionally different in these two experiments?

2. The Methods section needs a lot more detail and clarity, enough to allow anyone to replicate the study in exactly the same way.

3. The Discussion is missing many important points (details provided in the manuscript pdf)

We provide other minor comments, edits and suggestions directly in the manuscript pdf.

Reviewer #2 (Remarks to the Author):

The paper is very interesting and well written, with clear procedures. They also included limitations of the work, which is a well-come manner. The main conclusion is that redwings birds actively eavesdrop on and respond to yellow warbler seets as a frontline defense to protect their nest investment.

My main suggestion is to change the text in the statistical analysis topics. The analysis seems to be correct, but the analysis description should be more clear, specify the error distribution of the model, as in the case of GLM model. Where did you make the analysis? Wich post-hoc test did you made? Why it was suitable? Please, clarify that.

A minor consideration is with the word "seet". Please, consider defining the use of this term in the text.

Dear Editorial Board Members,

Thank you for your continued interest in our work. We have now revised the text according to the points of concern raised at this step of the review process. Please find our revised paper appended, and our point-by-point response letter inserted below.

We hope that our new version of the ms will be suitable for publication in *Communications Biology*.

Yours sincerely,
Shelby Lawson, corresponding author.

Reviewer #1 (Remarks to the Author):

Comment from reviewer:

1. The authors need to layout the logic of the study more clearly, in terms of:

- What pattern is expected in response to different treatments, to demonstrate that redwings are using warbler seet calls in a very specific way? This needs to be clearly spelt out in the Introduction and should guide the presentation of results and discussion.

Our response:

Overall, we predicted for experiment 1 that redwings should respond to yellow warbler seet calls (referential alarm call to cowbird) more so than to yellow warbler chip calls (general alarm call) (supported). In turn, for experiment 2 we predicted that redwings should show similar responses to cowbird chatters and yellow warbler seet calls (supported), but different responses to chatters and seets vs. blue jay calls (not supported). Our hypothesis and its predictions are made explicitly in the Introduction (line 98-116) and in the Discussion; we start with a paragraph about whether the overall findings support our expectations, and how each experiment builds on that.

Comment from reviewer:

- What is the motivation for doing experiments both in warbler territories and near redwing nests, i.e. what do the authors expect to be functionally different in these two experiments?

Our response:

This experiment started with playbacks on yellow warbler territories because we were initially only asking questions about yellow warbler behaviors. We were keeping track of other species' responses as well, and it was not until the end of that study that we discovered that not only the focal yellow warblers but also redwings (and not other species), responded to over half of the seet playbacks. To illustrate this point better, we now include a full data table on all the heterospecifics that responded to our different playback stimuli in experiment 1.

In turn, the next year we decided to conduct playbacks (experiment 2) focusing on redwing territories near their nests to test redwings' responses directly. We added two lines on line 99 to briefly describe this to readers: "The first experiment was conducted on yellow warbler territories for a separate study but that also comprised of heterospecific (including redwing) data. The second experiment sought to directly test redwings' responses to the playback types."

Comment from reviewer:

2. The Methods section needs a lot more detail and clarity, enough to allow anyone to replicate the study in exactly the same way (details provided in the manuscript pdf).

Our response:

Thank you for the suggestions! They were very helpful. We have made the changes and clarifications requested. Please see manuscript and comments below for specific changes.

Comment from reviewer:

3. The Discussion is missing many important points (details provided in the manuscript pdf).

Our response:

Thank you again for these detailed comments! Specific changes made are highlighted in the manuscript and described below.

Our responses to comments in the manuscript pdf:

***Line numbers are from updated manuscript, not the original**

Abstract

Line 26 – changed "usurp" to "eavesdrop on"

Introduction

Line 57 – referential calling is defined in line 52, but the sentence has been edited to make this even clearer

Line 78 – clarified sentence to say "The yellow warbler (*Setophaga petechia*) is the only known species to use referential alarm calls to signal the presence of a brood-parasite¹⁹. In response to brown-headed cowbirds (*Molothrus ater*, hereafter "cowbird")..."

Line 82 – Added line "Seet calls are only produced in response to cowbirds, and almost exclusively during laying and incubation when the nest is at the highest risk of parasitism"

Line 92 – Clarified sentence and moved reference to the end of sentence

Line 93 – Clarified frontload

Line 96 – changed "usurp" to "eavesdrop on"

Line 97 – Reviewer wrote "You need to explain what the different experiments are trying to test, i.e. what's the difference in doing the playbacks in warbler territories versus near cowbird nests?"

Our response:

We added two lines to line 99 explaining this. This experiment started with playbacks on yellow warbler territories because we were initially only asking questions about yellow warbler behaviors. We were keeping track of other species' responses as well, and it was not until the end of the study that we found that redwings responded to over half of the seet playbacks. The next year we decided to conduct playbacks on redwing territories to test their responses directly.

Line 107 – Reviewer wrote “I understand why you expect difference in response to seet calls for redwings nesting close/father away from warbler territories. But, for redwings who nest farther, why would you expect them to respond differently to cowbird chatters? This logic is unclear.”

Our response:

If redwings understood seet calls to always mean “cowbird”, then we expected the behavioral responses towards both calls to be similar. However, seet calls responses may be modulated depending on familiarity with and exposure/proximity to yellow warbler nests. That is why we grouped them for this particular analysis.

Methods

Line 129 – Added references

Line 132 – Information regarding average distance between redwing nests not available

Line 137 – A line about the specificity of the seet call was added to line 82 for a different comment. This has been tested extensively by Sharon Gill and Spencer Sealy. We added their references to line 137 as well.

Line 140 – Clarified why wood thrush is the control

Line 143 – Added line and reference for why blue jay call was added – “Using both a brood parasite and predator presentation is necessarily to fully understand whether the host's aggressive responses are specifically anti-parasitic or general”

Line 165 – Reviewer wrote “Could you provide a general measure of how far these territories were from each other?”

Our response:

There was a lot of variation in the distances between the different yellow warbler nests for experiment 1, and distances between the redwing nests in experiment 2. However, to make sure nests were not close enough to each other to have carryover effects or cause pseudoreplication, we set a minimum distance that nests had to be apart from each other to be given a playback based on average territory size of each species. This distance is specified on line 188 (experiment 1, >30m for yellow warbler territories) and 215 (experiment 2, >50m for redwing territories).

Line 190 – Reviewer wrote “Do you mean any given territory only received two of five possible treatments, with each treatment present on a different day? Please make this clear, and also

mention the gap, in days, between any two treatments at a particular territory. Mention if you randomized the order in which different treatments were played at each territory.”

Our response:

Thank you for the suggestion, this sentence has now been rewritten to be clearer about the playbacks “Yellow warbler territories received one of five different playback treatments on two separate days, such that each territory was tested with two of the five playback types”. The gap and randomized order are already mentioned on line 165.

Line 196 – Response variable was clarified “During the playback trials we specifically recorded a single binary response variable of whether any redwings responded to the playback or not. A response was only marked if one or more male redwings were present and alarm calling within 30 m of the speaker”

Line 198 – 30m was chosen because it is the radial average distance of territory size⁵⁴, this was added in the text as well

Line 217 – We chose nests over 50m apart to avoid resampling the same male twice. Nests at this distance are most likely different male “nest owners,” based on territory size of redwings (citation at end of sentence)

Line 225 – Added “instead of song posts to” to clarify

Line 229 – Reviewer wrote “This is not clear. Did you present all five treatments in each redwing territory here, while in the first experiment you presented only two treatments in each warbler territory? If yes, please explain the reason for this difference. “

Our response:

Thank you for pointing this out! The redwings were tested the same way as the yellow warblers, but we agree about the lack of clarity in the writing. We changed it to “Similar to experiment 1, redwings received one of five different playback treatments on two separate days, such that each pair was tested with two of the five playbacks”

Line 235 – Only male redwing responses were recorded in 2018 because only males responded to the playbacks on yellow warbler territories in experiment 1. After noting that during playbacks near redwing nests the female redwings responded as well, we recorded responses for both sexes in 2019. This sentence was added in the manuscript.

Line 235 – Reviewer wrote “Is it already known that these are “aggressive behaviours”?”

Response: Yes, they are used in other model presentation studies to rank aggressiveness of behaviors, both with redwings and other species.

Line 241 – Reviewer wrote “What is the significance of call rate in relation to this experiment?”

Our response:

Alarm calls are one of the main ways redwings show aggression which is why we recorded them. We recorded number of calls but changed it to a call rate metric just to simplify the numbers and the graphs.

Line 243 – Put call names in quotation marks as suggested

Line 250 – Replaced my sentence with yours as suggested

Line 253 – Reviewer wrote “How did you decide on 1 sec as the cutoff?”

Our response:

We broke it down like that because some of the playbacks had responses of 0 (as in birds reacted right when the playback started) while the control never had any responses of 0, and this was causing issues with our latency mixed models. So after consulting a statistician we decided to break the models into one for zeros (to see if any playback treatment had more zeros than others, meaning birds responded immediately more often to it than others) and another for non zeros (to see if disregarding birds that responded immediately, do we still see differences in latencies between treatments).

Line 253 – Reviewer wrote “What is a “zero response latency”? Is that different from “responded immediately (< 1 sec)”? Also, in the non-zero response latency analysis did you use number of seconds to respond as the dependent variable?”

Our response:

Thank you, we agree that this is confusing. We changed the sentence before this to say “We determined if redwing (males and females separately) responded immediately (latency of <1 sec) or with some latency (> 1 sec)”. The analysis was split so that we could test immediate responses (<1 sec) and non-zero responses (>1 sec) separately. We also added “(# of seconds to respond)” after “non-zero latency”.

Line 263 – Reviewer wrote “Why didn't you use other responses here?” for experiment 3.

Our response:

We only used call rate for experiment 3 because from the 2nd experiment it appeared that call rate provided the clearest distinction between treatments, thus we thought it best to use call rate as the response behavior for experiment 3.

Line 264 – Reviewer wrote “Can you be sure that there were no other yellow warbler territories at closer distances, that you might not have included in your experiment?”

Our response:

This is unlikely, because although yellow warbler nests can be cryptic, territorial males are highly detectably and their song is recognizable. Yellow warbler habitat is small and patchy at best at our sites; we are fairly confident we surveyed all territorial males as we thoroughly searched all potential yellow warbler habitat at all sites on multiple occasions. We likely did not find all nests at sites, but we likely found all the territories. A line about this was added on line 270.

Results

Line 286 – Reviewer wrote “Blue jay results have not been discussed here” under experiment 1.

Our response:

We only aimed to conduct 3 key comparisons in that dataset so that multiple comparisons did not interfere with significance testing. Because the playbacks on yellow warbler territories were initially done for another study (explained above), we have results for blue jay despite not having needed it for those first comparisons in experiment 1. We decided to leave these data in the presentations to allow the reader to make comparisons with experiment 2 directly.

Line 355 – Reviewer wrote “what is the expected response of cowbird chatters”?

Our response: We grouped responses to seet and chatter together for experiment 3 (reason listed in another comment above). We expected responses towards both to increase as distance from nearest yellow warbler nest decreased. We added “combined” after “seet and cowbird chatters” in text to clarify.

Discussion

Line 364 – Changed “Our results are consistent with patterns in which” to “Our results demonstrate a series of patterns in which” to make it clearer.

Line 373 – Did not discuss blue jay here for same reasons as comment above.

Line 383 – Added “neurofunctional research”

Line 384 – Added a line on 394 discussing response to redwing chatter “Redwings also responded to redwing chatter with the most aggression, suggesting that they perceive conspecific intruders as the gravest threat amongst our simulated set of danger cues.”

Line 384 – Rewrote sentence to clarify that the comparison is between the playbacks, not the sexes. The sexes did not differ. “In the second experiment, we found that when playbacks were presented on redwing territories, both males and females responded to seet calls with similar aggression (in terms of latency, alarm calling and closest approach) to cowbird chatters and blue jay calls, and both less so than to redwing chatters, indicating redwings treat seet calls with the same urgency as calls that simulate threats to breeding success, regardless of sex”

Line 389 – Reviewer wrote “Is this inference based on the fact that difference to seet calls and blue jay calls were not different?”

Our response:

Yes, we added this to the text to make it clearer – “Therefore, redwings appear to perceive seet calls as an indicator of the presence of nest threat, but not necessarily as a referential alarm call specifically informing brood parasitism threat, because responses to seets and Blue Jay calls did not differ”

Line 392 – Reviewer wrote “But you can't really say this just by the magnitude of the response. What you need to examine are the qualitative differences in behavioural response to different call types, i.e. is the behavioural response shown more similar between seet call treatment and cowbird chatter treatment when compared with seet call treatment versus blue jay call treatment.” in response to “The aggressive responses we observed in response to seet calls are likely a generic nest-protection response by redwings, as recorded by our observations.”

Our response:

We agree with the reviewers that rates of behaviors do not always measure the quality or intent of the measured behavior. For example, female redwings might approach conspecific male calls to mate but approach cowbird calls to attack. This is an inherent limitation of taking limited behavioral data. However, for our study we compared several types of behavioral responses (calling, approach, latency) and we still found the same patterns across behavioral responses metrics.

Line 395 – Reviewer wrote “But this doesn't explain the pattern in relation to cowbird calls - why should that be related to distance from warbler nests?”

Our response:

Changed sentence to include cowbird calls “These results suggest a “neighbourhood watch” effect, in that redwing males nesting closer to yellow warblers may be more sensitive to the latter species’ referential seet calls (and the presence of cowbirds through their calls), and as such redwings who have more access to the seet warning calls mount stronger frontline defenses to cowbirds and the seets compared to redwings that have access to this information.”

Line 407 – Added other species response table (Table 4) to the supplementary materials

Line 417 – Addressed reason for testing on yellow warbler and redwing territories in earlier comment

Line 422 – Addressed why we did not compare blue jays in the first experiment in an earlier comment

Line 439 – Earlier in the text we said that redwings have been found to reduce parasitism in the yellow warblers they nest near, likely because of their aggression towards cowbirds. We added a reiteration of this in the last paragraph to make it clearer: “Redwings that nest near yellow warblers respond more strongly to parasitic cues, and have been shown to reduce brood parasitism of the yellow warblers they nest near³⁷. Thus, it is possible yellow warblers and redwings have a mutualistic relationship where yellow warblers are the warning system for cowbirds and redwings are the mobbing aggressors that keep cowbirds at bay. “

Reviewer #2 (Remarks to the Author):

Reviewer wrote: The analysis seems to be correct, but the analysis description should be more clear, specify the error distribution of the model, as in the case of GLM model.

Our Response: The model distribution is on line 255, it is negative binomial.

Reviewer wrote: Where did you make the analysis?

Our response: We used R statistical program, version 3.6.1. (specified on line 167 in manuscript)

Reviewer wrote: Which post-hoc test did you made? Why it was suitable? Please, clarify that

Our response:

Thank you for pointing this out. We added that that the pairwise comparisons were “Tukey tests” in the text. We used Tukey tests as the standard post hoc test to control for multiple comparisons (because we had 5 playback treatments to compare with each other). We also clarified this on line 207 and 260.

Reviewer wrote: A minor consideration is with the word "seet". Please, consider defining the use of this term in the text.

Our response: Seet is what the call type has been termed since its discovery by Spencer Sealy and his lab group. We defined what it is used for in the text (line 79).

REVIEWERS' COMMENTS:

Reviewer #1 (Remarks to the Author):

The authors have done a very good job with the revision. We only have a couple of comments on this version.

1. In our earlier review we had commented: "I understand why you expect difference in response to seet calls for redwings nesting close/father away from warbler territories. But, for redwings who nest farther, why would you expect them to respond differently to cowbird chatters? This logic is unclear."

To which, the authors responded:

"If redwings understood seet calls to always mean "cowbird", then we expected the behavioral responses towards both calls to be similar. However, seet calls responses may be modulated depending on familiarity with and exposure/proximity to yellow warbler nests. That is why we grouped them for this particular analysis."

We are still not convinced by this, i.e. even if the functional meaning that redwings extract from seet calls is similar to cowbird calls, we still don't see why response to the latter should vary as a function of distance from warbler territory. The authors need to do a better job of justifying this, or change the way the handled this data.

2.The authors end their paper with the following sentence:

"Thus, it is possible yellow warblers and redwings have a mutualistic relationship where yellow warblers are the warning system for cowbirds and redwings are the mobbing aggressors that keep cowbirds at bay."

While this does sound like a great hypothesis, we think it is a bit of an overreach based just on these findings. The authors have presented no evidence to suggest that warbler calls are signals and not just cues and there is no discussion of redwing mobbing. Therefore, we suggest that the authors expand on this a little more and suggest this as a potential way foreard in understanding this system.

Reviewer #2 (Remarks to the Author):

The reviewers' suggestions were satisfactorily answered. It's a very interesting paper, addressing important insights.

Thank you.

Dear Editorial Board Members,

Thank you for your continued interest in our work (COMMSBIO-19-1757A). We have addressed the reviewer's final comments below. Please find attached our revised manuscript with the changes highlighted, and our point-by-point response letter inserted below.

Sincerely, Shelby Lawson and coauthors.

Reviewer #1:

The authors have done a very good job with the revision. We only have a couple of comments on this version.

1. In our earlier review we had commented: "I understand why you expect difference in response to seet calls for redwings nesting close/farther away from warbler territories. But, for redwings who nest farther, why would you expect them to respond differently to cowbird chatters? This logic is unclear."

To which, the authors responded:

"If redwings understood seet calls to always mean "cowbird", then we expected the behavioral responses towards both calls to be similar. However, seet calls responses may be modulated depending on familiarity with and exposure/proximity to yellow warbler nests. That is why we grouped them for this particular analysis."

We are still not convinced by this, i.e. even if the functional meaning that redwings extract from seet calls is similar to cowbird calls, we still don't see why response to the latter should vary as a function of distance from warbler territory. The authors need to do a better job of justifying this, or change the way they handled this data.

Our additional response:

To address this, we have added treatment (chatter vs. seet call playback) as a fixed effect for the analysis of variance. This new analysis reveals that calling responses of focal redwings show a similarly negative slope to both seets and to cowbird chatters in relation to the nest/territory distance from yellow warblers. Specifically, in the updated analysis, the treatment factor is not significant, indicating that the redwings' responses are not statistically significant between the playback types. Redwings males that nest near yellows may be more responsive to cowbird chatter than those that set up nests further away. We have made the following changes in the text:

Line 281- Updated methods to include treatment as an effect in analysis of variance

Line 357-363- Updated statistics results with treatment term included (not significant)

Line 405-407- Added a line to the discussion "Additionally, there was no difference between male redwings' own alarm responses to playbacks of seet vs. cowbird chatters, suggesting that redwings closer to yellow warblers are more responsive to both the warblers' referential calls and the cowbirds' own vocalizations than these calls reference."

2. The authors end their paper with the following sentence:

"Thus, it is possible yellow warblers and redwings have a mutualistic relationship where yellow warblers are the warning system for cowbirds and redwings are the mobbing aggressors that keep cowbirds at bay."

While this does sound like a great hypothesis, we think it is a bit of an overreach based just on these findings. The authors have presented no evidence to suggest that warbler calls are signals and not just cues and there is no discussion of redwing mobbing. Therefore, we suggest that the authors expand on this a little more and suggest this as a potential way forward in understanding this system.

Our response:

Thank you for the suggestion. We have reworded the last line to address and correct this overreach and we now suggest future research to explore a possible relationship between yellow warblers and redwings:

Line 448-452- "Given that redwing proximity has been shown to reduce the probability of brood parasitism by cowbirds upon yellow warbler nests nearby³⁷, our own findings open up questions for future research to explore whether yellow warblers and redwings possibly have a mutualistic communicative relationship, whereby yellow warblers are the alarm system provide warning cues for cowbird presence, and redwings are the aggressors that keep cowbirds at bay"

Reviewer #2:

The reviewers' suggestions were satisfactorily answered. It's a very interesting paper, addressing important insights.

Our response:

Thank you!